# rAAV Engineering for Capsid-Protein Enzyme Insertions and Mosaicism Reveals Resilience to Mutational, Structural and Thermal Perturbations

**DOI:** 10.3390/ijms20225702

**Published:** 2019-11-14

**Authors:** Rebecca C. Feiner, Julian Teschner, Kathrin E. Teschner, Marco T. Radukic, Tobias Baumann, Sven Hagen, Yvonne Hannappel, Niklas Biere, Dario Anselmetti, Katja M. Arndt, Kristian M. Müller

**Affiliations:** 1Cellular and Molecular Biotechnology, Faculty of Technology, Bielefeld University, 33615 Bielefeld, Germany; rebecca.feiner@uni-bielefeld.de (R.C.F.); julian.teschner@uni-bielefeld.de (J.T.); kathrin.schlicht@uni-bielefeld.de (K.E.T.); marco.radukic@uni-bielefeld.de (M.T.R.); 2Biocatalysis group, Department of Chemistry, Technische Universität Berlin, 10623 Berlin, Germany; tobias.baumann@tu-berlin.de; 3CO.DON AG, 10587 Berlin, Germany; sven.hagen@outlook.com; 4Physical and Biophysical Chemistry (PCIII), Department of Chemistry, Bielefeld University, 33615 Bielefeld, Germany; yvonne.hannappel@uni-bielefeld.de; 5Experimental Biophysics and Applied Nanoscience, Physics Department, Bielefeld University, 33615 Bielefeld, Germany; nbiere@physik.uni-bielefeld.de (N.B.); dario.anselmetti@physik.uni-bielefeld.de (D.A.); 6Molecular Biotechnology, Institute for Biochemistry and Biology, University of Potsdam, 14476 Potsdam, Germany; katja.arndt@uni-potsdam.de

**Keywords:** adeno-associated-virus, β-lactamase, inverted terminal repeat (ITR), loop modification, capsid stability

## Abstract

Recombinant adeno-associated viruses (rAAV) provide outstanding options for customization and superior capabilities for gene therapy. To access their full potential, facile genetic manipulation is pivotal, including capsid loop modifications. Therefore, we assessed capsid tolerance to modifications of the structural VP proteins in terms of stability and plasticity. Flexible glycine-serine linkers of increasing sizes were, at the genetic level, introduced into the 587 loop region of the VP proteins of serotype 2, the best studied AAV representative. Analyses of biological function and thermal stability with respect to genome release of viral particles revealed structural plasticity. In addition, insertion of the 29 kDa enzyme β-lactamase into the loop region was tested with a complete or a mosaic modification setting. For the mosaic approach, investigation of VP2 trans expression revealed that a Kozak sequence was required to prevent leaky scanning. Surprisingly, even the full capsid modification with β-lactamase allowed for the assembly of capsids with a concomitant increase in size. Enzyme activity assays revealed lactamase functionality for both rAAV variants, which demonstrates the structural robustness of this platform technology.

## 1. Introduction

Recombinant adeno-associated viruses (rAAV) are frequently used as a basic research tool and are emerging as therapeutic agents. For example, the US FDA recently approved voretigene neparvovec (Luxturna), which is based on rAAV serotype 2 (rAAV2) and delivers a gene to supplement biallelic RPE65 mutation-associated retinal dystrophy [1]. In addition, products based on AAV1 (Alipogene tiparvovec/Glybera) and very recently AAV9 (Onasemnogene abeparvovec/Zolgensma) obtained approval in Europe or the United States. The increasing impact of rAAV on gene therapy relies on a high safety profile resulting from the inability to replicate autonomously and on long-term target gene expression [2]. Wild-type AAVs have a non-enveloped, icosahedral capsid formed by 60 subunits of VP1, VP2 and VP3 proteins in an approximate molar ratio of 1:1:10 [3,4]. The regulatory mechanisms for expression of the three VP proteins is complex [5,6]. Briefly, splicing results in two mRNA transcripts that code for VP1 or for VP2/3, respectively. VP2 and VP3 are coded by the same mRNA and expression is controlled by a leaky-scanning mechanism at the VP2 start. In the AAV wild-type setting, the single-stranded DNA genome of about 4.7 kb includes two main reading frames (Rep and Cap) and is flanked by inverted terminal-repeat (ITR) sequences, which provide the encapsidation signal. Genetic engineering enables decoupling of the capsid coding genes from the encapsidated DNA. In the recombinant setting, *rep* and *cap* genes are provided in trans on a RepCap plasmid whereas a transgene expression cassette, frequently named gene of interest (GOI), is provided between the ITRs on the ITR plasmid. AAV needs additional ‘helper’ functionality from other viruses for production, which is provided on a separate pHelper plasmid [7,8]. For this reason, a three-plasmid system is often used, wherein the pHelper delivers the essential adenoviral elements E2A, E4 and the non-coding RNA VA. As host, HEK293 cells provide further adenoviral elements (E1A, E1B) and allow for high-titer production [7,8]. Alternative versions are also used such as a two-plasmid system combining the genetic information of the adenoviral helper sequences with AAV serotype specific *rep* and *cap* genes [9].

For diverse applications, e.g., virus-directed enzyme prodrug therapy (VDEPT) [10], viral targeting of specific cells is desired. AAV serotypes differ in their tropism and thus provide a first choice to achieve target specificity [11]. Deeper control over the target tropism requires genetic intervention. For this purpose, directed randomization and selection or rational engineering have been applied. Chimeric rAAV capsids are composed of proteins, which originate from different serotypes, and are often identified by evolutionary methods [12]. On the rational side, N-terminal fusions to e.g., the VP2 protein in rAAV2 have been studied [13,14,15]. In these cases, the addition of larger proteins, e.g., green fluorescent protein (GFP) and designed ankyrin repeat proteins (DARPins), was compatible with capsid assembly and targeting.

A further and early adopted rational approach, which is extended in this publication, is the integration of motifs in previously identified loop positions of the VP proteins. Two groups demonstrated that capsid formation and gene packaging are only slightly influenced by integration of peptide sequences in VP proteins at various residue positions [16,17]. Insertions in these positions have also been used for biorthogonal labelling of capsids [18,19]. Capsid accommodation capacity was shown for the integration of larger moieties such as the minimal F_c_-binding motif Z34C (34 amino acids) into the 587 loop region [20]. Production of such a Z34C rAAV2 with subsequent binding of an antibody was shown and transduction of target cells was observed. For vaccination via viral particle display, peptides up to 35 or 31 amino acids were integrated in the 453 or 587 position, respectively [21]. To our knowledge, the largest reported insertion to date is the fluorescent protein mCherry, which was functionally included in variable region IV at the 453 position of VP1, and allowed for the production of mosaic particles [22].

Our aim was to expand the loop modification strategy in combination with a systematic analysis of the engineering capacity of rAAV. For the construction of viruses, we extended an existing plasmid toolbox for rAAV2 manipulation and production [15,23]. Insertion of peptides in capsid proteins was studied with regard to rAAV productivity and transduction capability. The impact of capsid protein modifications on thermal stability has, to our knowledge, not been investigated. Thus, we first tested rAAV stability with glycine-serine insertions of varying length at residue position 587. These experiments confirmed that larger insertions are tolerated and we opted to insert the enzyme β-lactamase. As the introduction of an entire protein could interfere with capsid assembly, we tested partial insertions only in VP2 proteins. This required adaptation of the plasmid system for the production of mosaic rAAVs exclusively bearing VP2 loop modifications. Resulting mosaic particles were found to tolerate the insertion of a full-length β-lactamase in VP2 proteins. Finally, we set up a complete β-lactamase modification of all VP proteins. Production of these fully decorated rAAVs was possible and allowed for further characterization. In summary, our analyses demonstrated resilience of the virus to modifications at the genetic and protein level. We believe that the plasmid system combines facile genetic manipulation with a broad range of rAAV capsid engineering options.

## 2. Results

### 2.1. Modifications of ITR and RepCap Plasmids Are Compatible with rAAV Production

Despite the availability of different ITR and RepCap plasmids for rAAV2 production, options for facile loop capsid modification are sparse. In this work, previously modularized versions of the RepCap and ITR plasmids were used, which are largely compatible with the BioBrick RFC [10] cloning strategy (Appendix A) [23]. The RepCap plasmid (pZMB0216_Rep_VP123_453_587wt_ p5tataless) was retained but the ITR plasmid was redesigned to reduce cloning steps and serve user expectations outside the synthetic biology community. The new ITR plasmid (pZMB0522_ITR_EXS_ CMV_mVenus_hGHpA) contains the viral ITRs as part of a pUC19-based backbone, provides restriction sites for insertions based on BioBrick RFC [10] flanking the fluorescent reporter mVenus, and serves as a final destination plasmid. Plasmid generations are described in Method S1, final constructs used during this work are given in Table 1 and selected plasmids of the plasmid toolbox are given in Appendix A.

For sequence analysis of the ITRs, which commonly defy Sanger cycle sequencing, we cleaved the ITR DNA with BsaHI, generating two halves that are amenable to standard sequencing protocols (Appendix A, Method S2). Our plasmid pZMB0522 carries one complete ITR (in this plasmid notation referred to as 5′-ITR) and an ITR shortened by 11 bp (Figure 1).

Production of DNaseI-resistant particles and thus functionality of the ITR plasmid (pZMB0522) in combination with either the RepCap plasmid (pZMB0216) or a commercially available counterpart (pAAV-RC, GenBank: AF369963.1) was assayed using small-scale transfections and quantitative real-time PCR (qPCR). Genomic titers in crude lysates with both plasmids showed no significant differences (Figure 2a). A larger preparation (pZMB0522, pZMB0216) was purified by precipitation [25] and imaged by transmission electron microscopy. Capsid diameter measurements resulted in an average of 23.7 ± 1.2 nm (Appendix A), which is in good agreement with the expected value of 25 nm [24]. Manually counting over 500 particles yielded a fraction of filled capsid between 60% and 80% (Figure 2b, Appendix A).

To obtain samples with higher purity, all further preparations were purified by iodixanol gradient ultracentrifugation. For unmodified rAAV2 wt (pZMB0522, pZMB0216) genomic titers were determined by qPCR (Figure 2c). Using a comparable setup but depending on culture conditions and transfection efficiencies, titers between 1 × 10^10^ to 1 × 10^12^ vg/mL have been obtained.

Functionality of gene delivery was investigated with transduction assays. As common for AAV2, HT1080 cells were used, which express high levels of the rAAV2 primary receptor heparan sulfate proteoglycane (HSPG). Successful transduction was detected by the expression of the delivered transgene mVenus using flow cytometry. Based on a dilution series, the transducing titer was calculated. Genomes to infectious units yielded a specific infectivity of 16:1 (Figure 2c). This is in agreement with previous values, since for wild-type AAV2 a ratio of 1:1 and for rAAV2 ratios between 55:1 and 124:1 have been observed [26]. In addition, also transduction of the cancer cell lines A431, HeLa, MCF7, MDA-MB-231 and normal adult human dermal fibroblasts (HDFa) was tested with a multiplicity of infection (MOI) of 10,000 vector genomes per cell. Flow cytometry analysis (Figure 2d) revealed that the rAAV2 wt preparation was able to transduce a variety of different cells with high efficiencies. In agreement with previous reports, only the breast cancer cell line MCF7 showed low transduction [27]. These results demonstrated production and function of the rAAV plasmid system and provided the basis for further investigations regarding the tolerance of the viral capsid to insertions in the 587 loop region.

### 2.2. Systematic Variation of Loop Modifications Shows A Complex Pattern of Stability and Transduction Efficiency

Previous experiments showed that, VP proteins tolerate peptide insertions at residue positions 453 and 587 [16,17]. We aimed at systematically analyzing the impact of increasing insertion length on rAAV thermal stability and biological function. Since a protein had already been inserted in the 453 loop region [22] and the 587 loop region is even more frequently used for modification, we opted to integrate linkers with increasing size at residue position 587 and investigated production and function of the resulting rAAVs. Glycine-serine linkers were chosen because of their flexibility and solubility, resulting in incremental changes correlating with length. Specifically, amino-acid linkers with the sequences GG, GGSG, (GGSG)_2_ and (GGSG)_4_ were integrated at the genetic level in all VP proteins to yield homogeneously decorated viral particles as presented in a schematic overview in Figure 3a.

Cloning was facilitated by the unique restriction sites of RepCap plasmid pZMB0216 flanking the loop region (Appendix A). Genomic titers of iodixanol-purified preparations demonstrated that all genetic constructs lead to rAAV production and that integration of flexible linkers only affected the titer for large integrations (Table 2). rAAV transduction ability was determined based on mVenus expression in HT1080 cells incubated with a MOI of 50,000 viral genomes. This relatively high MOI was chosen to achieve reasonable signals above the detection limit despite the expected reduction of the transduction efficiency due to the capsid modifications. As for rAAV2 wt at these high MOIs the transduction level is highly saturated, lower MOIs were also analyzed. At a rAAV2 wt MOI of 5400 still 92%, at a MOI of 540 32%, and at a MOI of 54 3% of HT1080 cells were transduced. Increasing linker length impeded transduction (Table 2), but not in a linear fashion. The integration of only two amino acids resulted in a decrease of transduction ability from 97% (rAAV2 wt) to 31% (rAAV_587_GG) in the given experiment. Comparing the MOIs of rAAV2_587_GG and rAAV2 wt required to reach equal transduction of about 32%, the MOI of the mutant is even 93-fold higher. Interestingly, increasing the linker length to four amino acids (GGSG) resulted in a further decrease of transduction to 20%, but rAAV with an insertion of eight amino acids ((GGSG)_2_) showed a relatively improved transduction of 28% followed by a further drop in transduction to 11% for 16 amino acids (Figure 3b). Other MOIs ranging from 10 to 10,000 were also analyzed and overall yielded expected results. At MOI of 10,000 rAAV2_587_(GGSG)4 yielded 2.2% and at a MOI of 5000 1.7% positive cells (Appendix A).

Thermal stability of AAV capsids is an interesting biological and biophysical parameter and various methods to assess capsid stability have been described, such as differential scanning calorimetry (DSC), differential scanning fluorimetry (DSF) and electron microscopy [28], all of which monitor capsid breakdown but do not detect DNA release. We propose that the point of DNA release during heat treatment is a biological relevant event to describe capsid stability and that rAAV capsid integrity can be monitored by the DNase accessibility of the encapsidated DNA. Consequently, we repurposed the standard assay for genomic copy number determination and incubated rAAV samples at different temperatures before the treatment with DNase I. Subsequent analysis via qPCR yielded genomic copies of the rAAV sample plotted against the incubation temperature (Figure 3c–g). The disintegration temperature after a five-minute incubation (T_d, 5 min_) was determined as the temperature at which 50% of rAAVs have released their DNA. The term disintegration temperature T_d_ was chosen to distinguish the value from the melting temperature (T_m_) reported by other methods (e.g., DSF, DSC). The results for all glycine-serine linker variants are listed in Table 2. With an increase in linker size T_d_ decreases slightly, showing that the capsid is destabilized by large insertions, but also that despite significant structural intervention, stability is maintained at physiological temperatures.

To estimate the influence of the incubation time and the decay kinetics in our thermal release assay, rAAV2 wt was isothermally incubated at the previously determined T_d_ (56.1 °C) for different time points. The percentage of intact genomic copies decreased in a hyperbolic fashion albeit displaying different phases (Figure 3h).

### 2.3. Mosaic rAAVs with a 29 kDa β-Lactamase at Position 578 in VP2 Require a Kozak Consensus Sequence

To illuminate the possibly even larger insertion capacity of rAAV capsids, we chose the well-studied TEM β-lactamase [29] as a protein with a small distance between the N- and C-terminus, which approximately matches the distance of the β-hairpin residues in the variable region of the VP protein. The protein was inserted at the genetic level at residue position 587 of the VP proteins. Lactamase offers the possibility to easily measure enzyme activity. In order to avoid maximum interference with capsid assembly, β-lactamase was in this experiment incorporated only into the 587 loop of solely VP2.

The modular rAAV plasmid system, which we used as a starting point, had been deployed to produce N-terminal VP2 protein fusions and the respective modified rAAV particles [15]. A mutation in the RepCap plasmid eliminated the VP2 start codon (pZMB0600_Rep_VP13_453_587wt_p5tataless) and a fourth plasmid expressing VP2 and VP3 (here named shortly CMV_VP23 plasmid) was provided in trans [5,6]. For the previous modifications, the leaky scanning mechanism leading to the expression of VP3 did not pose a problem, because they were located at the unique N-terminus (Appendix A). However, since the 587 loop region lies within the coding sequence of both VP2 and VP3 proteins, loop modifications desired for VP2 only require suppression of concomitant expression of a likewise modified VP3.

In order to understand and insure the sole expression of VP2, we conducted an expression and mutation analysis starting with the RepCap and the CMV_VP23 plasmid. Transient transfection of HEK293 cells with this plasmid and subsequent Western blot analysis of VP expression was performed. Expression of all three VP proteins after transfection with the unmodified RepCap plasmid showed approximately the expected molar ratio between the three VP proteins (1:1:10) (Figure 4a, lane 1). A strong expression of both VP2 and VP3 proteins was observed for the CMV_VP23 plasmid (Figure 4a, lane 2). Apparently the leaky scanning mechanism is still active in the context of the CMV promoter and the cloning context (iGEM RFC [10]). To prevent undesired VP3 expression, a new plasmid abbreviated as CMV_VP2 was constructed, in which the VP3 start codon was removed by an exchange from ATG to ATC (Ile) (Figure 4b). As seen in the third lane of Figure 4a, expression of VP3 is unexpectedly still observed, which might be due to a second start codon located 24 bp downstream. To suppress leaky scanning, a strong Kozak sequence (GCC ACC) was introduced upstream of the VP2 start codon resulting in plasmid CMV_Kozak_VP2 (pZMB0315). Finally, solely the expression of VP2 (Figure 4a, lane 4) was detected with an expected increase in chemiluminescence intensity, indicating a higher level of expression.

The gene of the stabilized β-lactamase variant 14FM was cloned into plasmid CMV_Kozak_VP2. The resulting plasmid (pZMB0577_pSB1C3_001_CMV_Kozak_VP2 _453_587wtbla) was used for rAAV production in combination with Rep_VP13 plasmid (pZMB0600), the mVenus bearing ITR plasmid (pZMB0522) and pHelper. Two molar plasmid ratios of 5:5:1:4 and 5:5:4:1 (pHelper:ITR:Rep_VP13:CMV_VP2_587_bla) were tested for protein expression and production. Western blot analysis revealed a high proportion of VP2 proteins (Appendix A) for the 5:5:1:4 (higher amount of modified plasmid) ratio. Crude cell lysate samples from one 100 mm cell culture dish were analyzed regarding their genomic titers. The sample with the lower plasmid dose of the modified VP2_587_bla protein resulted in roughly three times higher amounts of viral particles (2.48 × 10^10^ vg/mL for a 1:4 VP13:VP2_587_bla ratio compared to 7.98 × 10^10^ vg/mL for a 4:1 VP13:VP2_587_bla (pZMB0600:pZMB0577)).

Based on the higher particle yield, mosaic rAAV2-VP2_587_bla particles were subsequently produced using the 5:5:4:1 ratio. A genomic titer of iodixanol-purified particles of 6.3 × 10^9^ vg was obtained per 100 mm dish. Thus, production using the four-plasmid system yielded rAAVs in the same range as the triple transfection production (Table 2). Incubation of HT1080 cells with a MOI of 50,000 resulted in about 57 ± 2% mVenus-positive cells (Table 2, Appendix A). Comparison with transduction values of rAAV2 wt shows a reduction of transduction ability for the enzyme-bearing particles (Appendix A). However, it should be noted that fully modified glycine-serine linker variants showed a much stronger reduction of the transduction ability.

We were interested if the enzymes presented on the capsid surface retained activity and thus, a colorimetric nitrocefin assay was performed. Enzymatic activity was evaluated from the linear correlation of absorbance against incubation time (Figure 4c). From this data we were able to estimate the number of active β-lactamases on the capsid surface. The β-lactamase variant 14FM which we used is a semi-rational combination of mutations described in literature (Hecky, Baumann unpublished data) [31]. Characteristics of this enzyme are presented in Appendix A. Combining the genomic copy number and the known turnover number of the free lactamase k_cat_ allows for the estimation of the total number of active lactamases in the sample. In this experiment a β-lactamase concentration of 1.73 × 10^−11^ mol L^−1^ was calculated. This value is equivalent to 5.6 enzymes per DNaseI-resistant particle. Such particles with five β-lactamases might look as illustrated in Figure 4d. In conclusion, mosaic rAAV2s (rAAV2-VP2_587_bla) with an incorporated full-length protein were produced with unaltered efficiency - thus capsid assembly was not posing a problem. Functionality for β-lactamase was proven for enzymes presented on the viral particle surface.

### 2.4. Fully Lactamase-Decorated rAAV Capsids Can Be Produced and Show Enzyme Activity

As a limit for the integration of motives into the capsid was not found in the mosaic experiment, we aimed at the production of rAAV with all VP proteins modified with a lactamase named rAAV2_587_bla. The β-lactamase gene was thus cloned in the RepCap plasmid (pZMB0216) at the 587 position (yielding pZMB0221_Rep_VP123_453_587bla_p5tataless). A fully modified capsid model is depicted in Figure 5a. The genomic titer after ultracentrifugation was comparable to those of glycine-serine linker insertions (Table 2). To our surprise, the insertion with the size of approximately 29 kDa in every VP protein did not abrogate capsid assembly.

We were interested to see how the 60 insertions would affect the overall size of the capsid. Measuring the particle width by atomic force microscopy (Figure 5b) a diameter of approx. 20 nm for rAAV2 wt was found. Note that the 5 nm deviation in particle size compared to the TEM measurements can be attributed to the type of method used and sample preparation. In contrast, a significantly higher viral particle diameter of approx. 29 nm was found for rAAV2_587_bla (Figure 5c). From the capsid model in Figure 5a a diameter of 35 nm was estimated for a fully enzyme-modified rAAV and estimations for the wild-type capsid resulted in an average of 25 nm, which agrees with the observation.

When rAAV2_587_bla samples were assayed for transduction ability on HT1080 cells, almost no transduction was measurable (Table 2). The thermal stability assay revealed a disintegration temperature T_d_ for rAAV2_587_bla of 55.6 °C. Surprisingly, this value corresponds approximately to that of rAAV2 wt. (Figure 5d, Table 2).

Besides testing the effect of the enzyme decoration on capsid integrity, functionality of the integrated β-lactamases was evaluated. From the nitrocefin assay a β-lactamase concentration of 1.65 × 10^−10^ mol L^−1^ was calculated, equivalent to 26.9 lactamases per DNaseI-resistant particle (Figure 5e). Looking from a different angle and assuming all 60 β-lactamase are active, yields a catalytic rate of 20,056.2 s^−1^ per capsid and a k_cat_ value of 334.3 s^−1^ for each β-lactamase. The presence of an active enzyme was furthermore studied in a simplistic bacterial growth assay. *Escherichia coli* cells lacking the corresponding resistance gene are not able to survive on agar plates supplemented with ampicillin. The presence of functional β-lactamase, however, leads to degradation of the antibiotic and thus bacterial growth is possible. Mixing samples of rAAV2 wt with *E. coli* does not allow for bacterial growth, whereas for samples of rAAV2_587_bla growth of bacterial colonies was observed (Figure 5f)—again indicating the presence of an active enzyme in the viral particle preparation.

## 3. Discussion

At the outset of the project, which relies on modular RepCap and ITR plasmids that are largely compatible with the BioBrick cloning standard, we first constructed an ITR plasmid.

In this context we reconsidered the very old problem, that the ITR sequences in plasmids are not stable during plasmid propagation in *E. coli* and that specifically one ITR is prone to deletions despite the symmetry of their sequences [32]. To our knowledge no explanation has been put forward. We hypothesize that the mechanism of plasmid replication of the prevalently used pMB1-derived origin of replication (a ColE1 type ori), which works unidirectionally with respect to RNA II elongation and includes a switch between PolI and PolIII about 200 bp downstream of the replication start, contributes to this genetic instability (Appendix A) [33]. In several ITR plasmids including our version, the distance between ori-start (end of the RNaseH digested RNAII transcript) and the ITR with the more frequent deletion is less than 100 bp. PolI and the polymerase switch, which possibly takes place around the observed deletion site, might be more prone to induce deletions compared to the PolIII mediated replication. Since the most often observed deletion of 11 bp is also present in commercial and Addgene plasmids (e.g., pAAV_MCS [34]) and since it has been reported that even larger ITR deletions reduce production but increase transgene expression [35], we maintained the deletion in one ITR and placed it strategically next to the ori. Due to the pUC numbering scheme and expected GOI orientation of our plasmid, the deletion is located in the right or 3’ ITR, respectively (Appendix A). Future experiments with various ITR to ori distances with high throughput ITR sequencing will test our hypothesis.

We were pleased to see that production of rAAV2 wt with the modified RepCap plasmid and the ITR plasmid (Table 2) enabled production in useful quantity and quality. Previous work has shown that insertion of motifs at various sites of VP proteins is possible [36,37]. We investigated glycine-serine linkers incorporated at residue position 587 by assaying capsid assembly, functionality and thermal stability. Production yields equal to that of rAAV wt showed compatibility with capsid assembly. Regarding transduction, we observed a significant impact of increasing insertion size with a non-linear decrease. The insertion of eight amino acids (GGSG)_2_ showed a significantly higher transduction compared to four amino acids (GGSG). Variable region VIII harbors residues R585 and R588, which mediate the primary interaction of AAV2 with the cell via HSPG [38,39]. Spatial separation of the two arginines is known to interfere with cell binding and internalization [40].

The insertion of small motifs presumably increases the tension within the β-hairpin and hence the distance between the arginines (Figure 6). Larger flexible insertions probably are compatible with the correct arginine positioning but with increasing size a shielding effect becomes more and more prevalent.

Next, we analyzed thermal stability using a qPCR-based method. We propose that the temperature of viral DNA release given by a disintegration temperature named T_d_ is a physical value that is on a par or even better corresponding to the biological activity of rAAV as values obtained by DSF or DSC experiments. It was hypothesized that viral particles transfer to a metastable state upon heating, which is defined by ejection of the encapsulated ssDNA [41,42]. Further temperature increases then completely rupture the capsid. DSC and DSF are more likely to capture the complete disassembly of the capsid, and thus are expected to result in higher T_m_ values compared to the T_d_ values. This is in agreement with our data which yielded disintegration temperatures about 15 °C lower than the melting temperatures published so far [28,43].

As seen in the time course of isothermal disintegration (Figure 3h), the particles are not in thermal equilibrium at elevated temperatures discouraging thermodynamic interpretations. Thus, the T_d_ values obtained are sensitive to the incubation time, which is expected for the megadalton complex. Interestingly, Figure 3h reveals a multi-phase behavior, which could be either interpreted by different composition or structural states of the initial capsid ensemble or by two unfolding pathways. Longer incubation times (>5 min) most likely result in even lower T_d_ values. Since most experimentally determined thermal stability changes within variants are minor, one can conclude that the capsid proteins form a stable framework highly resilient to sequence insertions.

Since peptide insertions of up to 16 amino acids were well tolerated, a β-lactamase was integrated at residue position 587. In one experiment β-lactamase was to be integrated only within the VP2 protein and not the VP3, which results from the same reading frame. An expression analysis of our existing CMV_VP2 plasmid revealed VP3 expression despite the CMV promoter for VP2 and even after deletion of the VP3 start codon. This can be explained by persistent leaky scanning and by a second AUG codon present 24 bp downstream of the mutated VP3 start. Mutating the first AUG of the VP3 protein was described before [44], but no Western blot was provided suggesting that a retained expression of VP3 could have occurred. That the VP2/3 gene retains expression of two proteins even if taken out of context and after a start codon removal indicates a deeply engrailed resilience to mutations and points to high robustness of the complex viral genome. We found that an additional strong Kozak sequence in front of VP3 resulted in the expression of solely modified VP2 [45].

The molar ratio of the VP protein expressing plasmids, needed to be controlled in order to provide the right amounts of each VP protein (1:1:10 for VP1:VP2:VP3) for desired capsid assembly and relative amount of modified VP protein. The AAV promoter p40 is weaker compared to CMV and the Kozak sequence additionally enhances expression. The ratio of 5:5:1:4 (pHelper:ITR:Rep_VP13:CMV_VP2_587_bla) was associated with a high proportion of VP2 protein, which potentially leads to a higher portion of modified VP2 in the assembled capsid compared to the wild-type ratio. Optimizing the quadruple setting, mosaic viral particles displaying lactamases were obtained with titers only slightly lower than those of wild-type preparations. Transduction efficiency of mosaic β-lactamase particles was reduced, but not as dramatically as in case of the 16 amino acids insertion into the homogenously modified rAAV_587_(GGSG)_4_. Most likely the lactamases pose a steric hindrance to the interaction with the cell, but the wild-type capsid proteins can still engage with HSPG. To our knowledge, this presents the first example of a successful enzyme incorporation at position 587 in the AAV capsid.

From a nitrocefin enzyme assay the number of active β-lactamases on the mosaic rAAV2_VP2_587bla capsid was estimated to be 5.6. Our calculations are based on the genomic titer and the proportion of empty capsids has not been taken into account. An alternative analysis of the full capsid titer using ELISA techniques is probably equally misleading as the modification of the capsid surface might interfere with antibody binding. Another uncertainty for the exact determination of the lactamase concentration lies within the assumption of a constant k_cat_ value, which was experimentally determined for the enzyme in solution. Due to immobilization on the capsids surface, the enzymes possess less degrees of freedom and substrate access is hindered, which might result in lower turnover numbers. Taken together, the true number of capsids is higher than estimated by genomic titer, because of empty capsids, but the true k_cat_ on the surface is lower than the one taken from solution measurements. It is conceivable that a heterogeneous mixture of active, partially active and inactive enzymes is presented, but due to the highly cooperative unfolding of lactamase this is less likely. Therefore, the deduced enzyme concentration should only be taken as an estimate. Still, the calculated number of 5.6 enzymes per viral capsid is very close to the expected number of modified VP2 proteins in the wild-type setting where five of the 60 capsid-forming proteins would be VP2. In combination, the data for a rAAV2 mCherry modification in the VP1 453 position [22] and for a lactamase modification in the VP2 587 position support the conclusion of a high structural plasticity of the AAV2 capsid.

In order to further explore the robustness of AAV assembly, the integration of β-lactamase was expanded to all VP proteins. We were intrigued to see that introduction of the enzyme into position 587 is possible while viral integrity and thermal stability are maintained. AFM measurements showed an increase in capsid diameter of 10 nm, which corresponds well to our estimations. Genomic titers were in the same range compared to the one of rAAV2 wt, which is remarkable considering the size of the β-lactamase. This suggests that VP folding, core capsid assembly and DNA encapsidation are not significantly affected by bulky insertions. Like the originating β-lactamase variants [31], the enzyme variant used herein is known to fold well and to possess high thermodynamic stability (Hecky, Baumann unpublished data), which might contribute to the robustness of the modified VP protein and the corresponding capsid assembly. The high thermal stability of the inserted enzyme might even protect the VP proteins from unfolding. In contrast, glycine-serine linkers are intrinsically disordered. As expected, they were found to destabilize the VP proteins and the overall capsid. Enzyme activity of the sterically confined β-lactamases was measured. Calculations on the basis of the genomic titer and the turnover number show that approximately the equivalent of 26.9 active enzymes are presented. As discussed above, enzyme activity could be reduced and deduced numbers of active enzymes per intact capsid present only an estimation.

In summary we provide plasmids for facile genetic manipulation of AAV2 capsids and illuminate the intriguing resilience of AAV2 to a breadth of genetic and structural modifications.

## 4. Materials and Methods

### 4.1. Construction of Plasmids

All constructs were made by standard cloning techniques mainly using idempotent cloning strategies according to RFC [10] or RFC [25], respectively [46,47]. AAV plasmids listed in Table 1 and Appendix A were cloned as described in Method S1. ITR sequencing is described in Method S2. The CMV and hGHpA containing plasmids are from the iGEM parts registry (parts.igem.org). Resulting vectors were analyzed for their correctness by Sanger DNA-sequencing (Sequencing Core Facility, CeBiTec, Bielefeld, Germany).

### 4.2. Cell Culture

HDFa (Thermo Fisher Scientific, Darmstadt, Germany), HEK293, HeLa, HT1080 (DSMZ) cells were cultured in Dulbecco’s Modified Eagle Medium supplemented 10% (*v*/*v*) fetal calf serum and 1% (*v*/*v*) penicillin/streptomycin (Sigma Aldrich, Steinheim, Germany). MCF7, A431 and MDA-MB-231 (DSMZ, Braunschweig, Germany) were cultured in RPMI supplemented 10% (*v*/*v*) fetal calf serum and 1% (*v*/*v*) penicillin/streptomycin. Cells were maintained at 37 °C and 5% CO_2_.

### 4.3. Viral Particle Production

HEK293 cells were seeded at a density of 3 × 10^6^ cells per 100 mm dish the day before transfection. A total amount of 15 µg DNA per 100 mm dish was transfected using calcium phosphate. RepCap plasmid, ITR-containing plasmid and pHelper plasmid were used in a 1:1:1 molar ratio [48]. For mosaic viral particles four plasmids were used in a 5:5:4:1 molar ratio of pHelper:ITR:RepCap_VP13:CMV_VP2. After 72 h of incubation at 37 °C, cells were harvested and pelleted by centrifugation (2000× *g*, 5 min).

### 4.4. Purification of Viral Particles

Cells were resuspended in lysis buffer (50 mM Tris, 150 mM NaCl, 2 mM MgCl_2_, pH 7.5) and viral particles were released from cells with three freeze-thaw cycles. Remaining DNA contamination was degraded by incubation with benzonase nuclease (final 100 U/mL, Sigma Aldrich) at 37 °C prior to addition of CHAPS (3-[(3-cholamidopropyl) dimethylammonio]-1-propane sulfonate, 0.5% w/v final). The crude lysate was cleared from cell debris by centrifugation (3000× *g*, 10 min). This crude viral stock was further purified with a discontinuous iodixanol gradient [49]. Briefly, the lysate was transferred onto a gradient of 60%, 40%, 25% and 15% iodixanol in an open top polyallomer 16 × 76 mm tube (Science Services, Munich, Germany). Tubes were sealed and centrifuged in a T-880 rotor (Sorvall) at 340,000× *g* for 2 h at 18 °C. The rAAV containing fraction was collected with a 21G x 1 1/2” injection needle and the buffer was exchanged to 1 × PBS (137 mM NaCl, 2.6 mM KCl, 10 mM Na_2_HPO_4_, 1.8 mM KH_2_PO_4_, pH 7.2) via Amicon Ultra-4 100K centrifugal filter units (Merck Millipore, Darmstadt, Germany).

### 4.5. SDS-PAGE and Western Blot Analysis

Cell pellets from rAAV production (1× 100 mm dish) were resuspended in 100 µL PBS and 5× SDS loading buffer. Samples were incubated at 95 °C for 10 min, centrifuged and 20 µL per lane were loaded on a 10% SDS-polyacrylamide gel (Hoefer SE260, Kleinbittersdorf, Germany). Samples were blotted onto a 0.45 µm nitrocellulose membrane (Thermo Fisher Scientific) using semi-dry electrophoretic transfer (V20-SDB, Scie-Plas, Cambridge, UK). After blocking the membrane with 10% (*w/v*) non-fat milk in TBS, the membrane was incubated simultaneously for 1.5 h with the B1 antibody (mouse monoclonal, supernatant, 1:100, Progen, Heidelberg, Germany) and an anti β-Actin antibody (8H10D10, mouse monoclonal, 1:1000, Cell Signaling Technology, Frankfurt am Main, Germany). After incubation with an anti-mouse IgG, HRP-linked antibody (1:5000, Cell Signaling Technology), blots were imaged by luminescence detection (Pierce ECL Western Blot Substrate, Thermo Fisher Scientific).

### 4.6. Determination of Genomic Titers

Before determination of genomic titers via qPCR, samples were treated with 10 U DNase I (New England Biolabs, Frankfurt am Main, Germany) in 10× DNaseI buffer in a final volume of 50 µL at 37 °C for 30 min before heat inactivation of the DNase I (75 °C, 20 min). Crude lysate samples were additionally incubated with 0.8 U Proteinase K (New England Biolabs) for 50 min at 37 °C before heat inactivation (95 °C, 10 min). Dilutions of the DNase I digest were used as template in the qPCR reaction. The sample was mixed with 2.5 µL primer qPCR-hGH-for (5′-CTCCCCAGTG CCTCTCCT-3′) and 2.5 µL primer qPCR-hGH-rev (5′-ACTTGCCCCT TGCTCCATAC-3′), each at a stock concentration of 4 µM, and 10 µL of 2 × GoTaq qPCR Mastermix (Promega, Mannheim, Germany). The qPCR reaction was carried out as described in the manual (TM318 6/14, Promega) with an increased time interval for the first denaturation step (95 °C, 10 min) using a LightCycler 480 II (Roche, Mannheim, Germany). The genomic titer was calculated from a standard curve of 10^2^ to 10^7^ copies of the ITR plasmid (pZMB0522) with an efficiency between 90% and 110% and an R value less than 0.1. Genomic titers in crude lysates were estimated from a standard curve mixed with the same amount of a non-transfected cell lysate.

### 4.7. Transducing Titer Assays

For determining transducing units (TU) 10,000 cells per well were seeded in 500 µL of the corresponding media on a 12-well plate, settled for 1 h at 37 °C and then ultracentrifugation-purified the respective rAAV sample was added in a serial dilution. After 12 h incubation, 500 µL fresh medium was added. After further 72 h incubation, cells were detached with 0.25% Trypsin/EDTA, resuspended in PBS and analyzed using a FACSCalibur and counting 10,000 events. The TU in mL^−1^ of rAAV stock solutions was calculated as an average of all diluted samples showing less than 40% of fluorescent-positive cells using formula: *TU* = *n* × *F_p_* × *d*/*V*, with *n* = number of cells in transduction experiment, *F_p_* = fraction of fluorescent cells, *d* = dilution factor of virus sample, and *V* = volume of virus sample in experiment. All experiments were performed as biological duplicates from two independent rAAV preparations. Transduction efficiencies at preset MOIs were determined based on the genomic titer using cultivation and measurements as described above. Histograms of flow cytometry analyses for transduction efficiency are given in Appendix A.

### 4.8. Transmission Electron Microscopy

Carbon-coated copper grids, 200 mesh (Science Services, Munich, Germany) were treated with oxygen plasma (Zepto, Diener electronic GmbH, Ebhausen, Germany). After this, 3 µL of precipitation-purified rAAV sample [25] was applied to the grid and incubated for 2 min. Excess liquid was drained off, the grid was dried at room temperature and washed with three drops of distilled water. Negative staining was performed using 3 µL 2% (*v*/*v*) uranyl acetate replacement stain (Science Services) for 30 s. Excess liquid was drained off and grids were dried before channeling the sample into the microscope. rAAVs were visualized with a CM100 (PW6021) instrument (Philips, Hamburg, Germany) with an acceleration voltage of 80 kV. Images were analyzed using the Soft Imaging Viewer (Olympus, Münster, Germany) and ImageJ [50].

### 4.9. Atomic Force Microscopy

AFM measurements of rAAV2 wt and rAAV2_587_bla were performed on a Multimode 8 AFM (Bruker, Karlsruhe, Germany) with Tap300Al-G cantilevers (BudgetSensors, Sofia, Bulgaria) in tapping mode in air. 2 µL of sample in PBS were spotted onto freshly cleaved mica and incubated for one minute. The mica was then briefly rinsed with water and dried under a gentle nitrogen flow. Data analysis was performed with Gwyddion 2.48. Obtained images were treated with offset and plane correction algorithms and the size of visualized particles was measured at half maximum particle height. Statistical analysis of size measurements was performed using Excel 2016.

### 4.10. AAV Stability Assay

Thermal stability was analyzed using iodixanol-purified rAAV2 variants in two technical replicates. rAAV2 samples were diluted to 1.5 × 10^9^ vg/mL with PBS. 10 µL aliquots of rAAV2 samples were incubated at temperatures ranging from 37.5 °C to 75.0 °C in 2.5 °C steps for 5 min in a thermocycler (peqSTAR 96 Universal Gradient, peqlab, Erlangen, Germany). To determine the disintegration time kinetics rAAV2 wt samples were incubated at 56.1 °C for different time points. Immediately after incubation, samples were stored on ice. Genomic titers of each sample (10 µL) were determined by qPCR as described above. Due to the conceptual identity to the qPCR titration method, sensitivity and reliability are from this point on well established. A complete digest of the not encapsidated DNA was assured by additional tests. We evaluated the minimum titer of the starting material for a successful evaluation of vector particle stability. This needed to be higher than 1 × 10^9^ genomic copies per mL in order to result in a data set that covers the sigmoidal-shaped disintegration curve. Our qPCR standard curve shows a linear correlation between 10^2^ and 10^7^ genomic copies per sample which represents the dynamic range. Note that our qPCR primers hybridize in the hGHpolyA region and that we detect completeness of this region. Probing of different parts or the whole genome is possible. Data analysis was performed using Origin2018. The total rAAV amount per reaction was normalized to 100% and a non-linear regression with the logistic function $y=A_{min}+\frac{\left(A_{max}-A_{\min}\right)}{{\left(1+{\left(\frac{x_{0}}{x}\right)}^{h}\right)}^{s}}\;$ was performed with *A_min_* = 0, *A_max_* = 100, no weighting. The disintegration temperature (T_d_) was determined as the temperature at which 50% of rAAVs released its DNA.

### 4.11. β-Lactamase Activity Assays

Activity of β-lactamase presented on the rAAV capsid was determined using a nitrocefin and a bacterial assay. For the spectrophotometric nitrocefin assay 91 µL of rAAV sample containing either rAAV2_587_bla or rAAV2_VP2_587_bla (each 3.7 × 10^8^ vg total) or PBS (negative control) were mixed with 9 µL of nitrocefin buffer (2 mM nitrocefin, 500 mM KH_2_PO_4_ and 5% (*v*/*v*) DMSO at pH 7.0) in a 96-well plate. Absorption at 486 nm was measured with a microplate spectrophotometer (PowerWave HT, BioTek, Bad Friedrichshall, Germany) in 1 min intervals for 40 min at room temperature. Calculations are based on the Lambert–Beer rule as given in the formula: $\frac{\Delta A}{\Delta t}=\;{\Delta \varepsilon }_{486}\cdot c_{bla}\cdot k_{cat_{bla}}$ with $\frac{\Delta A}{\Delta t}=$ slope of linear regression from nitrocefin assay; ${\Delta \varepsilon }_{486}=$ molar extinction coefficient of nitrocefin shift (here 16000 l mol^−1^ cm^−1^); $c_{bla}=$ concentration of β-lactamase; $k_{cat_{bla}}=$ catalytic constant of β-lactamase; d= distance (here 0.267 cm). Either the turnover number (here 746 s^−1^) or the concentration of active lactamases was assumed to be known.

In the bacterial assay an aliquot of about 200 µL of an *E. coli* DH5α culture transfected with plasmid pSB1C3_BBa_J04450 harboring a constitutively expressed chloramphenicol acetyltransferase and a red fluorescent protein was plated on LB agar supplemented with ampicillin (100 µg mL^−1^ final) and chloramphenicol (20 µg mL^−1^ final). rAAV samples (3 µL) were spotted (rAAV2_587_bla undiluted, 1:2, 1:3 and rAAV2 wt undiluted). The dish was incubated at 37 °C overnight. Due to lacking ampicillin resistance, growth and colony formation of *E. coli* cells only occur, when ampicillin is hydrolyzed by active β-lactamase of the rAAV sample. Growing *E. coli* colonies could be easily detected by eye via their red appearance.

### 4.12. Statistical Analysis and Reproducibility

Standard deviation was calculated for all biological and technical replicates. To test whether differences were statistically significant based on a 0.05 significance level, data were checked for a normal distribution by a Shapiro–Wilk test, then an independent Student’s *t*-test was performed and *p*-values are given according to: * *p* ≤ 0.05; ** *p* ≤ 0.01; *** *p* ≤ 0.001. Variation of transduction efficiency determination was exemplarily tested in fully independent experiments with independent viral rAAV2 wt preparations by two authors. These data agree nicely with 95.7% (Figure 2d, Appendix A) and 96.7% (Table 2, Appendix A). Variation in the qPCR stability assay was tested with two independent viral preparations in two settings as shown in Figure 3c,h. In both assays, 50% of genomic copies are detected after 5 min incubation.

## Figures and Tables

**Figure 1 ijms-20-05702-f001:**
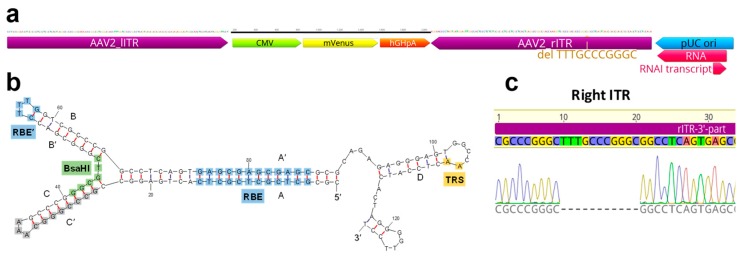
Overview of the inverted terminal repeat (ITR) sequences of rAAV2. (**a**) Scheme of ITR plasmid components. The ITRs are enlarged compared to the interchangeable gene of interest (GOI) (CMV promoter, mVenus fluorescent protein gene and hGHpolyA sequence). The deleted 11 bp sequence is highlighted in the right ITR adjacent to the pUC ori. (**b**) Structure of one ITR computed by Mfold webserver (150 mM NaCl, 5 mM MgCl_2_, 37 °C) [24]. Standard Sanger DNA-sequencing of ITRs was enabled by digestion with BsaHI whose recognition site is highlighted. (**c**) Results of Sanger-DNA sequencing aligned to original ITR sequences. Fragments were sequenced with oligonucleotides given in Method S2. The 3′-part of rITR shows a deletion of 11 bp (5′-TTTGCCCGGGC-3′). Sequencing results of the remaining ITR fragments are shown in Appendix Ac,d.

**Figure 2 ijms-20-05702-f002:**
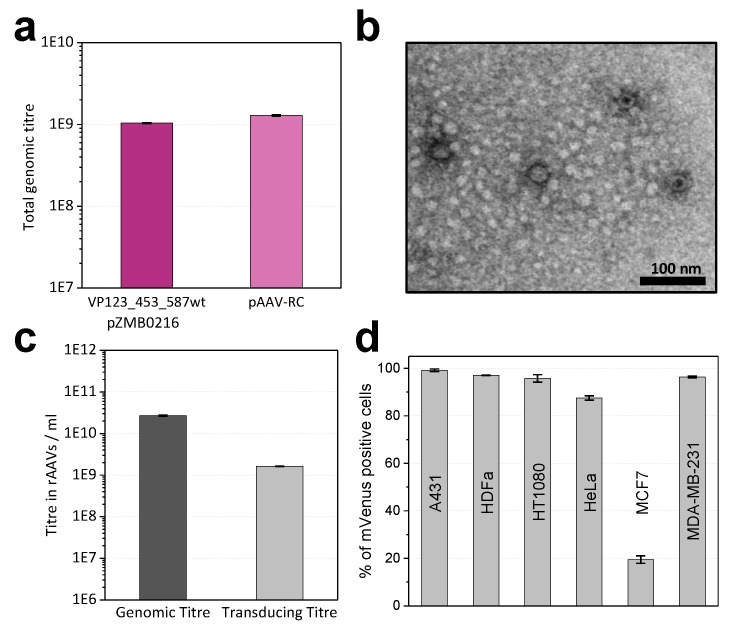
Production of recombinant adeno-associated viruses (rAAVs) packaging the fluorescent reporter mVenus gene with plasmids pZMB0522 and pZMB0216. (**a**) Production yields of rAAV samples determined by qPCR from crude cell lysate from one 100 mm cell culture dish. Comparable values were obtained either using the new RepCap plasmid (left bar) or the commercial pAAV-RC plasmid (right bar). Standard deviations of three biological and two technical replicates were calculated for each sample type. (**b**) Transmission electron microscopy image analysis of precipitation-purified viral samples at 39,000-fold magnification revealed viral particles with a size of about 24 nm. (**c**) Comparison of viral genomic and transducing titers. Genomic titers were determined via qPCR from three technical replicates of ultracentrifugation-purified viral samples. Transducing titers were analyzed by mVenus expression from biological and technical duplicates after flow cytometry analysis of HT1080 cells. (**d**) Transduction of different cell lines. Cells were incubated with a multiplicity of infection (MOI) of 10,000. Biological duplicates at two time points were analyzed by flow cytometry, measuring mVenus fluorescence and counting 10,000 events. Histograms of flow cytometry analysis can be found in Appendix A.

**Figure 3 ijms-20-05702-f003:**
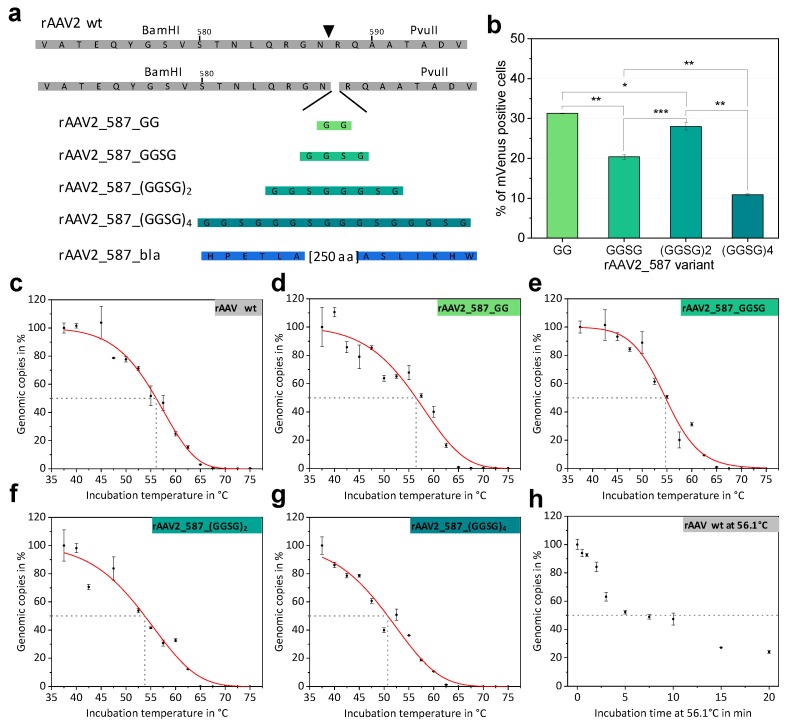
Comparison of different 587 loop variants and results of thermal stability assays. (**a**) Schematic overview on the coding sequence of the rAAV wt VP sequence. Restrictions site positions enabling the introduction of motifs flanking the 587 position (triangle) are indicated and amino-acid sequences of inserted linkers are shown. For β-lactamase only the first and the last amino acids are given. (**b**) Transduction efficiencies of different serine-glycine linker rAAV variants. Statistical analysis was performed as described in the Material and Methods section and *p*-values are given according to: * *p* ≤ 0.05; ** *p* ≤ 0.01; *** *p* ≤ 0.001. (**c**–**g**) Thermal stability assays of rAAV particles measured in PBS. The percentage of qPCR detected genomic copies is plotted against the incubation temperature. Each point represents a technical duplicate with the standard deviation. Fitted curves (red) were calculated using a logistic function to determine the disintegration temperature for all rAAV variants. Table 2 lists the T_d_ values. (**h**) Disintegration kinetics of rAAV2 wt samples incubated at T_d_ = 56.1 °C for different time periods. A dashed line indicates the point where 50% of genomic copies were released from the capsid.

**Figure 4 ijms-20-05702-f004:**
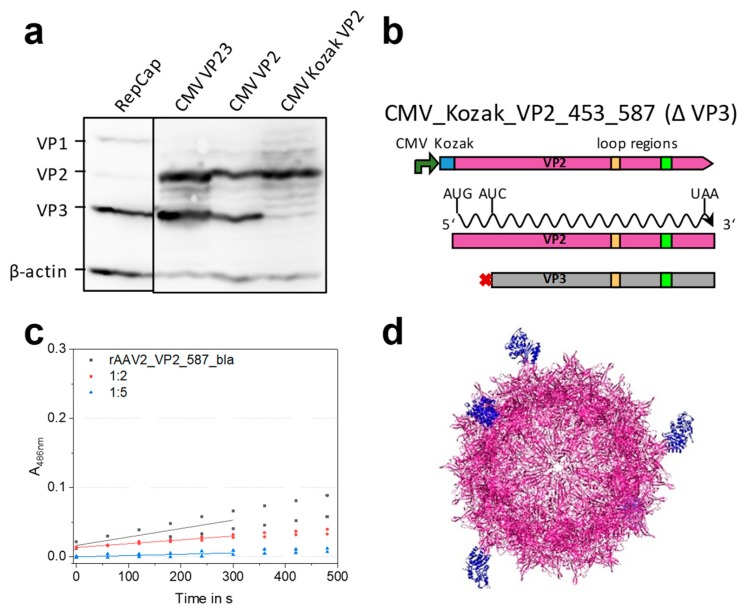
Establishment of mosaic 587 rAAV production using a modified four-plasmid system. (**a**) Analysis of *cap* protein expression after transient transfection in Western blots of crude cell lysates. Transfection of pZM0216_RepCap showed the expected ratio of 1:1:10 for VP1, VP2 and VP3 (lane 1). Three plasmid versions for VP2 expression were analyzed containing (i) the VP2 and VP3 cassette (lane 2, CMV_VP23, pZMB0160), (ii) the cassette with VP3 start codon removal (lane 3, CMV_VP2, pZMB0298), and (iii) the cassette with an upstream Kozak sequence and VP3 start knock out (lane 4, CMV_Kozak_VP2 (pZMB0315). Expression of VP3 after removal of the start codon can be explained with a second start codon 24 bp downstream and persistent leaky scanning. Appendix A shows full length images. (**b**) Scheme of the final expression construct. (**c**) Nitrocefin assay probing rAAV2_VP2_587_bla mosaic viral particles in different concentrations for lactamase activity. Slopes of linear regressions of the first 300 s were used to calculate the concentration of active β-lactamases. (**d**) Theoretical structure of the mosaic particle (UCSF Chimera [30]) composed of the AAV2 wt structure (purple, PDB: 1LP3) and five copies of β-lactamase (blue, PDB: 3DTM).

**Figure 5 ijms-20-05702-f005:**
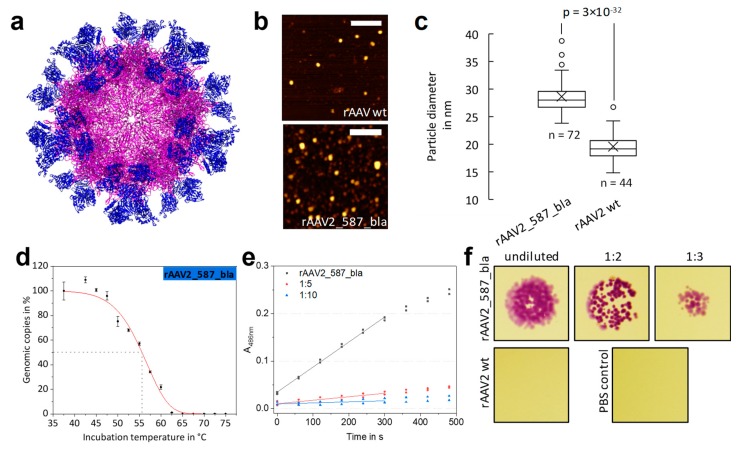
Comparison of rAAV2 wt and rAAV2_587_bla with respect to structure and β-lactamase activity. (**a**) Theoretical structure of rAAV2_587_bla constructed from the AAV2 wt structure (purple, PDB: 1LP3) and 60 copies of β-lactamase (blue, PDB: 3DTM). (**b**,**c**) AFM micrographs and particle size analysis of rAAV2 wt and rAAV2_587_bla. The calculated mean diameter of rAAV2 wt is 20 nm and 29 nm for rAAV_587_bla. Appendix A gives AFM raw data. (**d**) Results of thermal stability assay based on qPCR data for rAAV_587_bla. Each point represents the standard deviation of a technical duplicate. T_d,5 min_ was calculated with 55.6 ± 0.4 °C. (**e**) Nitrocefin assay, probing the β-lactamase activity of rAAV2_587_bla in different dilutions. (f) Images of a bacterial growth assay on LB Agar plates for β-lactamase activity. Samples with rAAV2_587_bla show a β-lactamase activity up to 1:3 dilutions, whereas for rAAV2 wt and PBS no colony growth was observed.

**Figure 6 ijms-20-05702-f006:**
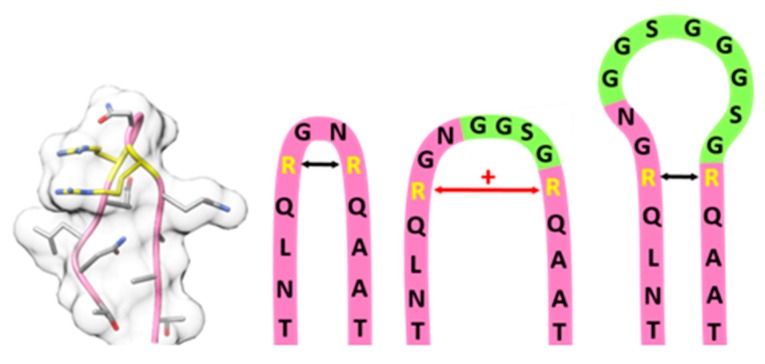
Model of the 587 loop region, from left to right: Structural model of the wild-type 587 loop with residue R585 and R588 highlighted in yellow (PDB: 1LP3); schematic model of 587 wild-type loop region (pink), 587 loop region with GGSG linker (light green) insertion leading to an increased arginine-arginine-distance; 587 loop region with (GGSG)_2_ linker (light green) insertion leading to a regular arginine-arginine-distance but steric shielding of heparan sulfate proteoglycane (HSPG) binding motif.

**Table 1 ijms-20-05702-t001:** Overview of plasmids submitted to Addgene including a short description of their features. Information on the cloning of the plasmids is available in the Appendix A.

Plasmid Name with Description	Length	Backbone
pZMB0522_ITR_EXS_CMV_mVenus_hGHpA AAV2 ITR flanking a CMV promoter expressing the fluorescent protein mVenus	4014 bp	pUC19
pZMB0216_Rep_VP123_453_587wt_p5tataless expression of VP1/2/3 of AAV2 with cloning ready 453 and 587 loop regions, arginines in 587 loop region are intact, p5 promoter at end of expression cassette	5455 bp	pSB1C3_001
pZMB0600_Rep_VP13_453_587wt_p5tataless expression of VP1 and VP3 with cloning ready 453 and 587 loop regions Arg in 587 loop region are intact	6455 bp	pSB1C3_001
pZMB0315_CMV_Kozak_VP2_453_587wtHis expression of VP2 with Kozak sequence to prevent leaky scanning and VP3 start knock out, cloning ready 453 and 587 loop regions, Arg in 587 intact, His-tag in 587 loop	4705 bp	pSB1C3_001

**Table 2 ijms-20-05702-t002:** Genomic titers, transduction efficiency and disintegration temperatures for wt and modified rAAV variants.

Sample	Titer in vg/mL ^a^	Transduction Ability in % ^b^	T_d, 5 min_ in °C^c^
rAAV2 wt	3.1 × 10^10^	96.7 ± 0.1	56.1 ± 0.5
rAAV2_587_GG	7.1 × 10^9^	31.3 ± 0.1	56.4 ± 0.8
rAAV2_587_GGSG	4.0 × 10^10^	20.4 ± 0.6	54.7 ± 0.5
rAAV2_587_(GGSG)2	2.4 × 10^10^	28.0 ± 0.9	53.8 ± 0.8
rAAV2_587_(GGSG)4	4.7 × 10^9^	10.9 ± 0.2	50.7 ± 0.7
rAAV2_587_bla	1.3 × 10^10^	1.2 ± 0.1	55.6 ± 0.4
rAAV2_VP2_587_bla	6.3 × 10^10^	57.0 ± 2	n.d.

^a^ Genomic titers are given in viral genomes (vg) per mL as determined by qPCR. Each value corresponds to a production with 10 × 10 cm cell culture dishes and a final purified volume of 0.5 mL. ^b^ Transduction ability was assayed with flow cytometry of HT1080 cells using a MOI of 50,000 and is given as percentage of mVenus expressing cells. The error is based on biological triplicates from one viral preparation. ^c^ Disintegration temperatures T_d_ were determined in the qPCR-based stability assay.

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
