# Peer review of "rAAV Engineering for Capsid-Protein Enzyme Insertions and Mosaicism Reveals Resilience to Mutational, Structural and Thermal Perturbations"

_ijms, 2019, doi:10.3390/ijms20225702_

Round 1

Reviewer 1 Report

Review of Manuscript “rAAV Engineering for Capsid-Protein Enzyme Insertions and Mosaicism Reveals Resilience to Mutational, Structural and Thermal Perturbations” by Feiner et al..

The authors describe the generation of adeno-associated virus serotype 2 based recombinant vectors with insertions in the amino acid 587 loop of the capsid (VP) proteins. They could show that thermal stability of the corresponding capsids was mostly preserved after introduction of glycine-serine linkers of variable length or the complete ß-lactamase gene and present plasmid-based systems for the generation of such capsid-altered rAAV2 vectors with no major loss of packaging efficiency as compared with the wt capsids. They could show that ß-lactamase activity was preserved in the context of the AAV capsids.

The manuscript is written quite comprehensible. However, many issues in the results and discussions are described with background informations and technical details, which should be rather relocated to the introduction, materials or supplementary sections to improve the flow of reading (e.g description of the regulation of AAV-2 capsid protein expression in lines 242 to 247 or generation of VP2_587bla mosaic virus particles in lines 279 to 294). In summary the manuscript is written in a rather technical fashion.

One major point of criticism is that the transductions properties of the genetically modified variants via the regular AAV2-wt entry and uncoating mechanisms may be reduced far more than presented in the paper. Thus the conformation of the capsids may be grossly altered already after insertion of short peptid sequences, at least if all three capsid protein VP1, VP2 and VP3 are modified. This shortcoming is explained in more detail below as major point 1.   

Detailed list of major and minor points:

1) The transduction ability as presented by the percentage of transgene positive HT1080 cells after transduction with very high MOIs (50,000, probably GP/cell) of the wt capsid and capsid modified rAAVs is a very poor parameter for comparison (lines 195 to 202, and table 2). First, the percentage of transgene positive cells is not at all a linear function of the number of input viral particles. With higher MOIs the increases in the percentage of transgene positive cells tend to level and reach saturation. At an MOI of 50,000 the percentage of transgene positive cells is highly saturated (96,7%) for the wildtype rAAV, which can also be seen by comparison with Fig. 2d, where a similar very high percentage of transgene positive cells was obtained with a MOI of 10,000. Thus the transducing titers of the rAAV variants may actually be several decimal powers lower than that of the wildtype.  

Therefore the transducing titers obtained from dilution series should be presented for the wt and the variant rAAVs. For this it is also necessary to provide exact information on how the transducing titer were calculated from these dilution experiments (e.g extrapolation of compensation curves) in the Materials section (see also minor point 2 below).

2) The rather lengthy first section of the discussion section deals with aspects (maintenance of AAV ITR sequences in bacteria) rather unrelated to the scope of the present paper and should be omitted.

Minor points:

1) Fig. 1 C is barely readable in a print-out and should either be enlarged or omitted (and presented in the supplementary information)

2) Transduction assays with purified rAAV preparations, lines 152 to 165: It should be stated whether the MOI of 10,000 used for these experiments refers to genomic or transducing particles per cell.

Furthermore, under the conditions employed the transduction of all cell lines used except for the MCF7 cells is already in the saturation range. Therefore, to be able to compare the other cell lines, experiments should be repeated with lower MOIs.

It should also be described in more detail in the Methods section (lines 530 to 535), how exactly the titer of transducing particles was determined from the dilution series (line 155)

Author Response

The manuscript is written quite comprehensible. However, many issues in the results and discussions are described with background informations and technical details, which should be rather relocated to the introduction, materials or supplementary sections to improve the flow of reading (e.g description of the regulation of AAV-2 capsid protein expression in lines 242 to 247 or generation of VP2_587bla mosaic virus particles in lines 279 to 294). In summary the manuscript is written in a rather technical fashion.

We agree with the reviewer and shifted the mentioned text passages either to the introduction or to the discussion section.

Therefore, the transducing titers obtained from dilution series should be presented for the wt and the variant rAAVs. For this it is also necessary to provide exact information on how the transducing titer were calculated from these dilution experiments (e.g extrapolation of compensation curves) in the Materials section (see also minor point 2 below).

Indeed, we did not explain the rationale behind the chosen high MOIs. Since the introduction of a full protein in the capsid shell is a major modification, we expected a drastically reduced transduction efficiency compared to that of the minor modifications. To fit this wide range of transduction efficiencies in one consistent experimental setup and maintain a focus on our new variants, we opted for the high MOI. As the reviewer correctly points out this might saturate the observable transduction efficiency of wt rAAV. As requested, we carried out additional experiments and titrated MOIs. These data are now available in the supplement. In addition, we explain the high MOI in the main text.

2) The rather lengthy first section of the discussion section deals with aspects (maintenance of AAV ITR sequences in bacteria) rather unrelated to the scope of the present paper and should be omitted.

Since we feel that our manuscript also covers the plasmids necessary to produce modified rAAV, we think that this discussion is related to the topic. The paragraph in question has only 259 words and for now we kept it. ITR stability is a major issue for rAAV production.

Minor points:

1) Fig. 1 C is barely readable in a print-out and should either be enlarged or omitted (and presented in the supplementary information)

We enlarged Fig. 1c after cropping part of the chromatogram and focusing on the deletion.

2) Transduction assays with purified rAAV preparations, lines 152 to 165: It should be stated whether the MOI of 10,000 used for these experiments refers to genomic or transducing particles per cell.

We apologize for not making this clear in our manuscript. We used the genomic titer for further calculations and included this information in the manuscript.

Furthermore, under the conditions employed the transduction of all cell lines used except for the MCF7 cells is already in the saturation range. Therefore, to be able to compare the other cell lines, experiments should be repeated with lower MOIs.

We acknowledge that our values might be in the saturation range but chose this experimental setup to enable a comparison with a previous publication from another group, which even used a MOI of 100 000 (!) [Ellis, B. L.; Hirsch, M. L.; Barker, J. C.; Connelly, J. P.; Steininger, R. J.; Porteus, M. H. A Survey of Ex Vivo/in Vitro Transduction Efficiency of Mammalian Primary Cells and Cell Lines with Nine Natural Adeno-Associated Virus (AAV1-9) and One Engineered Adeno-Associated Virus Serotype. Virol. J. 2013, 10 (1), 74.]. We cite the publication at the respective text section. Browsing the literature reveals that many research groups use high MOIs, when they want to compare a broader range of capsid modifications or cells. Our data support our conclusion that the virus particles function as expected in comparison with the literature.

It should also be described in more detail in the Methods section (lines 530 to 535), how exactly the titer of transducing particles was determined from the dilution series (line 155)

We added a more detailed description in the Methods section to improve understandability and reproducibility.

Reviewer 2 Report

Feiner et al., describe an elaborate research work on engineering of rAAV.

Pros:

Well thought out design of experimentation. Presented nicely.

Cons:

1B: The particles to the far right seem to show distinct nucleic acid material in the (dense) center compared to the two on the left (they seem to be empty particles). What percent of the purified virus is infectious? What is the transduction efficiency? Why have the authors used such high MOI of 50,000? What is the data if the MOI is say 10, 100, and 1000?

Author Response

1B: The particles to the far right seem to show distinct nucleic acid material in the (dense) center compared to the two on the left (they seem to be empty particles).

Figure 2B shows two empty and two full viral particles. Due to the negative staining procedure the particles on the right with the dense core are empty. This interpretation is used in many publications.

What percent of the purified virus is infectious? What is the transduction efficiency? Why have the authors used such high MOI of 50,000? What is the data if the MOI is say 10, 100, and 1000?

Indeed, we did not explain the rationale behind the chosen high MOIs. Since the introduction of a full protein in the capsid shell is a major modification, we expected a drastically reduced transduction efficiency compared to that of the minor modifications. To fit this wide range of transduction efficiencies in one consistent experimental setup and maintain a focus on our new variants, we opted for the high MOI. As the reviewer correctly points out this might saturate the observable transduction efficiency of wt rAAV. As requested, we carried out additional experiments and titrated MOIs. These data are now available in the supplement. In addition, we explain the high MOI in the main text.

Reviewer 3 Report

The manuscript entitled “rAAV Engineering for Capsid-Protein Enzyme Insertions and Mosaicism Reveals Resilience to Mutational, Structural and Thermal Perturbations” by Feiner et al. assessed AAV2 capsid tolerance to insertions into the 587 loop region. These insertions consisted in four glycine-serine linkers and the 29 kDa enzyme β-lactamase. Insertion of β-lactamase was tested with a complete or mosaic (inserted in VP2 only) modification setting. All the insertions were tolerated and capsid stability is maintained at physiological temperatures. The β-lactamase presented on the capsid surface retained activity.

Recombinant adeno-associated viruses (rAAV) are emerging as therapeutic agents for gene therapy, with several of AAV-based clinical trials currently ongoing. There is a vast record on the literature of genetic modification of rAAV in order to achieve a desired tropism, improve delivery to target cells, production efficiency, evasion of pre-existing immunity and to increase packaging capacity. One of the common modifications consist in the integration of motifs in previously identified loop positions in VP proteins. This work extend this rational approach, providing the first example of a successful enzyme incorporation at position 587 in the AAV capsid.

The authors successfully apply and combine a wide set of experimental approaches, including molecular biology, enzymology, cell culture, protein biochemistry, transmission electron microscopy and atomic force microscopy. They also developed a qPCR based AAV stability assay. The results are interpreted appropriately and all conclusions justified and supported by the results. The manuscript is well written, in appropriated English. The results would be of interest for the whole AAV field.

Despite what was said above there are some minor issues that need to be addressed to improve the overall quality of the manuscript:

Minor issues:

Capsid diameter measurements are done with two different techniques, transmission electron microscopy and atomic force microscopy. TEM is used only for WT capsids and AFM for the capsids with insertions. Why this? Why TEM and/or AFM are not used in all the cases?

The use of transmission electron microscopy allows to quantify the fraction of filled capsids, as shown in Figure 2 and S3. How does the insertions affects the fraction of filled capsids?

The Td, 5 min value for rAAV2_VP2_587_bla is not reported. Was it determined? If no, why?

Line 409: Correct typo, replace “Figure 2h” by “Figure 3h”.

Author Response

Minor issues:

Capsid diameter measurements are done with two different techniques, transmission electron microscopy and atomic force microscopy. TEM is used only for WT capsids and AFM for the capsids with insertions. Why this? Why TEM and/or AFM are not used in all the cases?

The reviewer is right that it would have been nice to compare all data with the same method. We had technical problems with the TEM and the instrument ultimately was out of order for an unspecified time period. For this reason, we switched to AFM.

The use of transmission electron microscopy allows to quantify the fraction of filled capsids, as shown in Figure 2 and S3. How does the insertions affects the fraction of filled capsids?

Due to the broken TEM, we have not been able to analyze all variants for empty and full capsids and thus, were not able to determine this really interesting value for all samples.

The Td, 5 min value for rAAV2_VP2_587_bla is not reported. Was it determined? If no, why?

The Td, 5 min value for rAAV2_VP2_587_bla was not determined as it requires high quantities of material and since the effect of even more lactamase insertions in rAAV2_587_bla did show only a minor change in stability.

Line 409: Correct typo, replace “Figure 2h” by “Figure 3h”.

The numbering was corrected.

Round 2

Reviewer 1 Report

Review of Revised Version Manuscript “rAAV Engineering for Capsid-Protein Enzyme Insertions and Mosaicism Reveals Resilience to Mutational, Structural and Thermal Perturbations” by Feiner et al..

Most of the points raised in the initial review have been addressed in a proper fashion in the revised version of the manuscript.

However, for one of the major points, which is cited below, the response of the authors and the changes made in the manuscript are not quite satisfactory:

“The transduction ability as presented by the percentage of transgene positive HT1080 cells after transduction with very high MOIs (50,000, probably GP/cell) of the wt capsid and capsid modified rAAVs is a very poor parameter for comparison (lines 195 to 202, and table 2). First, the percentage of transgene positive cells is not at all a linear function of the number of input viral particles. With higher MOIs the increases in the percentage of transgene positive cells tend to level and reach saturation. At an MOI of 50,000 the percentage of transgene positive cells is highly saturated (96,7%) for the wildtype rAAV, which can also be seen by comparison with Fig. 2d, where a similar very high percentage of transgene positive cells was obtained with a MOI of 10,000. Thus the transducing titers of the rAAV variants may actually be several decimal powers lower than that of the wildtype.  

Therefore the transducing titers obtained from dilution series should be presented for the wt and the variant rAAVs. For this it is also necessary to provide exact information on how the transducing titer were calculated from these dilution experiments (e.g extrapolation of compensation curves) in the Materials section (see also minor point 2 below).“

Response:

Indeed, we did not explain the rationale behind the chosen high MOIs. Since the introduction of a full protein in the capsid shell is a major modification, we expected a drastically reduced transduction efficiency compared to that of the minor modifications. To fit this wide range of transduction efficiencies in one consistent experimental setup and maintain a focus on our new variants, we opted for the high MOI. As the reviewer correctly points out this might saturate the observable transduction efficiency of wt rAAV. As requested, we carried out additional experiments and titrated MOIs. These data are now available in the supplement. In addition, we explain the high MOI in the main text.

A consistent experimental setup is not necessarily limited to one defined (in this case very high) MOI of the wt rAAV and the rAAV variants, but may include MOIs over a wide range, so that it assured that the values used for direct comparison in the text are neither in the saturation range not below the detection limit. For example, at present it is still stated in the text that “the integration of only two amino acids resulted in a decrease of transduction ability from 97% (rAAV2 wt) to 31% (rAAV_587_GG) in the given experiment“, which implies a moderate reduction in transduction efficiency of a factor of only 3 to 4. As shown in fig. 2c, the wt rAAV has a genomic to transducing particle ration of about 16. So for the wt rAAV, the MOI of genomic particles per cell to reach 31% positive cells would calculate to about 5 genomic particles (0.31 x 16) per cell. From these values the amounts of the wt and the 587_GG needed to obtain the same percentage of transduced cells differ by a factor of 10000 (MOI 5 vs. MOI 50000) in this specific example. Whereas the authors have now included the transduction ability of some of their mutants at low MOIs of GP/cell as S1 Table 2, the most important controls, the wt rAAVs at low MOIs, are still missing. As a minimum requirement, these wt rAAV values (which must have been obtained for the graph in Fig. 2c), should be included into the table and the table should be part of the main text, so that the interested reader is able to directly estimate the differencies in transduction efficiencies on her/his own.

Minor point: line 663: “?? =n × Fp × d /V“ should read “??/ml =n × Fp × d /V“.  

Author Response

A consistent experimental setup is not necessarily limited to one defined (in this case very high) MOI of the wt rAAV and the rAAV variants, but may include MOIs over a wide range, so that it assured that the values used for direct comparison in the text are neither in the saturation range not below the detection limit. For example, at present it is still stated in the text that “the integration of only two amino acids resulted in a decrease of transduction ability from 97% (rAAV2 wt) to 31% (rAAV_587_GG) in the given experiment“, which implies a moderate reduction in transduction efficiency of a factor of only 3 to 4. As shown in fig. 2c, the wt rAAV has a genomic to transducing particle ration of about 16. So for the wt rAAV, the MOI of genomic particles per cell to reach 31% positive cells would calculate to about 5 genomic particles (0.31 x 16) per cell. From these values the amounts of the wt and the 587_GG needed to obtain the same percentage of transduced cells differ by a factor of 10000 (MOI 5 vs. MOI 50000) in this specific example. Whereas the authors have now included the transduction ability of some of their mutants at low MOIs of GP/cell as S1 Table 2, the most important controls, the wt rAAVs at low MOIs, are still missing. As a minimum requirement, these wt rAAV values (which must have been obtained for the graph in Fig. 2c), should be included into the table and the table should be part of the main text, so that the interested reader is able to directly estimate the differencies in transduction efficiencies on her/his own.

We added information on the transduction efficiency of the rAAV2 wt and give a comparison to the rAAV2_587_GG at the 32% transduction elvel. The main text now reads:

"As for rAAV2 wt at these high MOIs the transduction level is highly saturated, lower MOIs were also analyzed. At a rAAV2 wt MOI of 5400 still 92 %, at a MOI of 540 32 %, and at a MOI of 54 3 % of HT1080 cells were transduced. Increasing linker length impeded transduction (Table 2), but not in a linear fashion. The integration of only two amino acids resulted in a decrease of transduction ability from 97% (rAAV2 wt) to 31% (rAAV_587_GG) in the given experiment. Comparing the MOIs of rAAV2_587_GG and rAAV2 wt required to reach equal transduction of about 32 %, the MOI of the mutant is even 93-fold higher."

Minor point: line 663: “?? =n × Fp × d /V“ should read “??/ml =n × Fp × d /V“.

In our understanding, the equation given in the text is a 'quantity equation' and according to SI rules such equations do not include units. We clarified the text accompanying the equation. It now reads:

The TU in ml-1 was calculated as an average of all samples showing less than 40 % percent of fluorescent-positive cells using following formula: ?? = n × Fp × d / V, with n = number of cells in transduction experiment, Fp = fraction of fluorescent cells, d = dilution factor of virus sample, and V = volume of virus sample in experiment.